# Effect of Vertical and Horizontal Sample Orientations on Uniformity of Microwave Heating Produced by Magnetron and Solid-State Generators

**DOI:** 10.3390/foods10091986

**Published:** 2021-08-25

**Authors:** Somayeh Taghian Dinani, Alina Jenn, Ulrich Kulozik

**Affiliations:** Chair of Food and Bioprocess Engineering, TUM School of Life Sciences, Technical University of Munich, 85354 Freising, Germany; alina.jenn@tum.de (A.J.); ulrich.kulozik@tum.de (U.K.)

**Keywords:** vertical and horizontal sample orientations, magnetron and solid-state microwave systems, heating patterns, Maillard browning reaction, infrared pictures

## Abstract

In this study, the effect of different horizontal and vertical orientations of a model sample (cuboid gellan gel samples containing Maillard reactants) on microwave heat processing was investigated in the solid-state and magnetron microwave systems. To achieve this target, seven orientations inside both microwave cavities were defined. Two of the investigated sample orientations were in a vertical position with and without turntable rotation, and five in a horizontal position. Furthermore, samples at horizontal orientations were put at an angle position without turntable rotation. To analyze the microwave heating patterns, infrared (IR) pictures and photographs of the gellan gel samples were taken after processing to document IR-based thermal and Maillard color changes, respectively. Three main factors for improvement of the heating homogeneity were identified: first, processing samples in the solid-state microwave system; second, position variation of the sample by turntable activated; and third, horizontal orientation. In addition, it was observed that placing the gellan gel samples in a vertical position in the magnetron microwave system resulted in considerably more absorbed power and a more uniform microwave heat processing compared to other horizontal orientations in this system. This indicated a non-uniform microwave field distribution. The results of this study can also confirm the importance of designing suitable food packaging: a vertical shape for more microwave energy absorbance and thus, more energy efficiency, and a horizontal shape for more uniform microwave heat processing.

## 1. Introduction

Microwave heating offers an easy, fast, and convenient way of preparing food for consumers [1,2]. In addition, it has several other advantages such as prompt electric control, energy efficiency, and non-pollution of the environment [3] due to the use of clean energy [4]. Therefore, microwave heating is extensively used on a domestic scale and is also gradually used in industrial applications [3,5]. The applications of microwave heating in the food processing sector can be various including reheating, tempering, defrosting, cooking, baking, pasteurization, sterilization, drying, etc. [2].

Although microwave heating in principle provides a volumetric and, thus, a more uniform heat processing compared to conventional heating by conduction or convection [2], one of the main limitations of microwave heating is non-uniform temperature distribution of the processed sample [6]. This limitation can result in overheat processing in hot spots and, simultaneously, possibly unsafe heat processing in cold spots of processed foods. This issue is at least partially related to an uneven distribution of the electromagnetic field inside the microwave cavity [2]. Therefore, although microwave heat processing has an irreplaceable position in households, it has so far not been exploited to its fullest potential in industrial applications, especially in the food industry [7]. In fact, to actually achieve uniform volumetric microwave heating is an ideal aim and therefore the subject of ongoing research and development in the food industry [5].

One promising technology is the application of solid-state microwave generators instead of magnetrons [1,8,9,10,11]. In solid-state microwave technology, the initial phase of the varying field can be controlled at any electromagnetic inlet port and, thus, the position of the nodes and antinodes can be changed in a targeted manner. Moreover, the power of any port can be changed independently to increase the number of electric field patterns in the solid-state microwave system. In this system, the frequency can be changed dynamically to change the resonant mode number [5,12,13]. Finally, although magnetron microwave systems can only be switched on/off for several seconds during heat processing, providing a coarse control, solid-state microwave systems can provide both coarse and fine control (switching on/off for microsecond), which makes the power delivery much more linear and controllable in this system [10]. Therefore, solid-state microwave systems can potentially replace traditional magnetron microwave systems in the near future [9,14].

Moreover, shape [15], position, and orientation of the samples are factors to improve microwave heating uniformity [11]. In the paper by Taghian Dinani, Hasić, Auer, and Kulozik (2020) [16], we compared three different shapes of samples including spherical, cylindrical, and cuboid shapes. In this paper, the cuboid shape was suggested as the best, perfectly showing the changes of microwave field distribution. Taghian Dinani, Kubbutat, and Kulozik (2020) [11] compared two positions of cuboid samples in both magnetron and solid-state systems: at the center and near the edge of the turntable. However, the orientation of cuboid samples in the center of the turntable has not been investigated until now. Considering sample placement inside the microwave cavity, in particular, vertical and horizontal orientation can be effective and should be investigated. By changing the orientation of that sample from vertical to horizontal, not only is the sample surface in physical contact with the turntable, but also the sample surfaces exposed to the microwaves are changed. In this case, the microwave field distribution may be changed inside the microwave cavity. In addition, by changing the orientation of the sample from vertical to horizontal and by the selection of different horizontal orientations, the position of the sample relative to the microwave port is changed. Therefore, it can be effective for microwave absorption and microwave field distribution in the sample. Therefore, the goal of this study was to investigate the effects of the vertical and horizontal orientation of samples and orientation variants in terms of the angle relative to the microwave inlet port and changing positions in the cavity on microwave heat uniformity. Differences between samples processed in the magnetron and the solid-state microwave system were compared. To the best of our knowledge, a study combining all these factors in comparing different vertical and horizontal orientations in both magnetron and solid-state microwave systems is lacking. As a model sample, gellan gel cubes containing Maillard reaction reactants were used. The famous Maillard reaction is a non-enzymatic browning process that occurs between amino acids and reducing sugars [17]. The brown color formation during the Maillard reaction can be used and measured for three-dimensional visualization of the heating pattern of samples during microwave processing [18,19]. In other words, the application of the time-temperature mapping technique of the Maillard chemical marker method is an appropriate tool to monitor microwave heat uniformity inside the microwave heated samples for the optimization and controlling of microwave processes [20,21,22,23]. Moreover, this method offers a reliable and fast visual and optical procedure to detect the position of cold and hot spots in the heated sample [11,24]. To analyze the heating patterns of samples, Maillard reaction-related color changes and temperature distribution measured by an infrared camera were used.

## 2. Materials and Methods

### 2.1. Identifying Parallel Output Power Levels in Each Orientation for Microwave Heat Processing in Both Microwave Generators

Since the primary goal was to compare the effect of different sample orientations on the heat uniformity of the model food systems, it was important to expose them to a similar level of absorbed power in similar orientations in both solid-state and magnetron microwave systems. In other words, for an impartial comparison of each orientation in both microwave systems, the equivalent output power levels for both microwave systems in each orientation should be adjusted to lead to similar levels of absorbed power in the samples [11]. In this study, the output power level of 300 W was picked for all samples processed in different orientations in the magnetron microwave system. Therefore, corresponding parallel output power levels for each orientation were identified in the solid-state microwave system to obtain the same absorbed powers in both microwave systems, as explained in the following. 

To measure the power absorbed by samples in both magnetron and solid-state microwave systems, double deionized (DDI) water was used as a model system to easily measure the temperature increase. Although the characteristics of DDI water such as its dielectric properties are different in comparison to the solid gellan gel samples, a liquid sample such as DDI water needs to be used to determine the absorbed power levels in the microwave systems. It is related to the fact that it is necessary to mix the sample very quickly after microwave heat processing to have an even temperature distribution in the whole sample, and thus provide the opportunity to conveniently measure the sample temperature for the calculation of absorbed microwave energy by the sample. Therefore, the energy absorption by samples at different orientations and in both microwave systems can be determined with negligible errors and very precisely to have a fair comparison of different orientations and microwave systems using this procedure [11]. In more detail, two Teflon beakers with the same shape and internal dimensions representing vertical and horizontal gellan gel samples were prepared (Figure 1). Then, they were filled with 58 g of DDI water with a temperature between 0 °C and 10 °C. This weight equals the weight of one gellan gel cuboid sample with dimensions of 3.2 cm × 3.2 cm × 6 cm with the described formula and production procedure in Section 2.2. It is worth mentioning that the DDI water was previously cooled down to prevent excessive evaporation during microwave heat processing for 120 s. This time duration was similar to the microwave heat processing time of the gellan gel samples in different orientations.

Next, the output power levels of 100 W, 300 W, and 440 W in the magnetron microwave system and 50 W, 150 W, 250 W, and 300 W in the solid-state microwave system were applied for each orientation. It must be noted that in the magnetron microwave system, other power levels than 300 W such as 100 W and 440 W were investigated to establish a linear relationship between the absorbed power and output power levels in this microwave system. To measure the absorbed power, the temperature of DDI water inside the Teflon beaker was measured after fine water stirring right before and after microwave heating with a thermocouple (Testo 108, Testo SE & CO, Titisee-Neustadt, Germany). Then, the absorbed power (Pabs) was calculated with this equation [10,11,25]:(1)$$P_{\mathrm{abs}}=\frac{{m\;\times \;c}_{p}\;\times \;\Delta T}{\Delta t}$$

In Equation (1), m is the weight of the DDI water sample (58 g); c_p_ is the specific heat capacity of water (4.187 J g^−1^ K^−1^ at ambient pressure); ΔT is the temperature difference between the start and end of the heat processing; and Δt is the duration of microwave heat processing (120 s). For each orientation and power level, three replications were performed.

In the next step, the mean values of three replications of absorbed power (*y*-axis) were plotted against the output power (*x*-axis) in the solid-state microwave system for each orientation and the linear equations were calculated. Then, the absorbed power of the magnetron system at the output power of 300 W in a similar orientation was inserted in the related linear equation to calculate the adjusted output power in the solid-state system. Therefore, the calculated parallel output power levels in the solid-state microwave system for each orientation resulted in a similar absorbed power level as the magnetron microwave system at the output power level of 300 W. All equations and calculated parallel output power levels of different sample orientations are summarized in Table 1.

### 2.2. Preparation of Gellan Gel Cuboids

To prepare gellan gel samples, the formulation and preparation methods of our previous paper [20] were used. In more detail, the liquid mixture containing l-lysine (2%), d-ribose (2%), gellan gum (1.5%), calcium chloride dehydrate (0.59%), anatase titanium dioxide (0.01%), and double deionized (DDI) water (93.9%) was prepared. This formulation was selected after several trials. It is important to mention that in designing a suitable model food system, it is crucial to find the optimal concentrations of initial Maillard substrates as well as other ingredients to provide a detectable increment in Maillard color changes of the processed sample during or after microwave heat processing [18]. Furthermore, it is necessary to prevent color changes at saturated color changes in the whole sample. The prepared gel solution with the temperature of 77 °C was quickly poured into cubical molds with internal cross-sections of 3.2 cm × 3.2 cm, which were placed in an ice bath to make the cooling process faster and prevent the Maillard color changes of the gels before the microwave heat processing. When the gels cooled down completely, they were taken out of the molds and stored at the temperature of 10 °C overnight. To process the gels the next day, the gels were cut into 6 cm high pieces before microwave heat processing.

### 2.3. Microwave Heat Processing of Gellan Gel Cuboids

For microwave heat processing, the gels were taken from the cooling storage room. Then, they were cut to a length of 6 cm. At the end, each sample was 6.0 cm long, 3.2 cm wide, and 3.2 cm high. For the microwave heat processing of samples, they were placed on round Teflon plates (Figure 2) with a cavity custom-sized to fit the gels. The Teflon is a non-dielectric substance [26] and it was used to prevent heat conduction between the heated gellan gel sample and the turntable and thus provide better recognition of the heated sample in the infrared (IR) pictures. For microwave heat processing of the samples, the cavity was 6.0 cm long and 3.2 cm wide for all horizontal positions, while it was a square with 3.2 cm length for both vertical positions. The cavities of both Teflon plates were 2 mm deep. These cavities were designed to put samples on the Teflon plates at a similar place and at their center the whole time (Figure 2). The advantage of using Teflon plates for microwave heat processing of samples is that they are almost insulated from microwaves and thus are not considerably affected by microwave heating. Therefore, they make it easier to distinguish between the sample and surrounding area in the thermal images.

Then, the gellan gel cuboid samples were processed for 120 s at a frequency of 2.45 GHz with magnetron (Panasonic NN-SD681S, Panasonic, Japan) and solid-state (FM–SSMWG–500 W 2.45 GHz, Fricke und Mallah Microwave Technology GmbH, Peine, Germany) microwave generators using the described output power levels for each orientation (see Table 1). It is worth mentioning that both magnetron and solid-state microwave cavities were fragmented according to the four cardinal directions (0° as north, 90° as east, 180 as south, and 270 as west) to be able to name each of the seven investigated orientations. These seven orientations are displayed in Figure 3 and the following abbreviations were used to distinguish them:H–0°–OFF: Horizontal orientation in north–south direction without turntable rotation.H–135°–OFF: Horizontal orientation in northwest–southeast direction without turntable rotation.H–45°–OFF: Horizontal orientation in southwest–northeast direction without turntable rotation.H–90°–OFF: Horizontal orientation in west–east direction without turntable rotation.H–ON: Horizontal orientation with turntable rotation.V–OFF: Vertical orientation without turntable rotation.V–ON: Vertical orientation with turntable rotation.

**Figure 3 foods-10-01986-f003:**
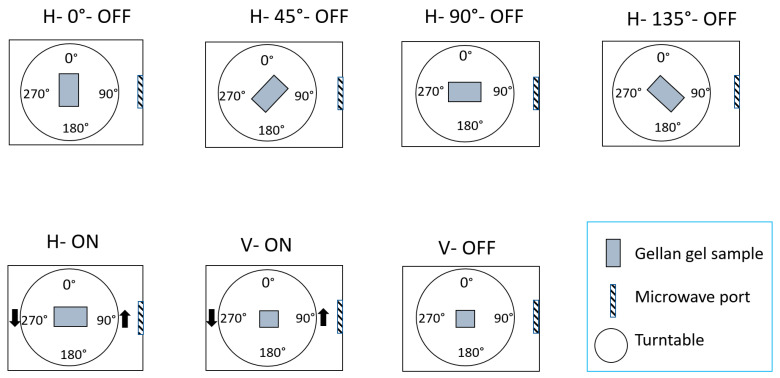
Seven investigated orientations in both the solid-state and magnetron microwave cavities. The abbreviations of V, H, ON, and OFF represent vertical orientation, horizontal orientation, turntable on, and turntable off, respectively.

After microwave heat processing, the heated samples were quickly removed from the microwave cavity and cut into four even slices with the same thickness of 8 mm using a bread-cutting machine (E16, Ritterwerk GmbH, Sendling, Germany). Figure 4 shows that each slice was labeled with a letter and each surface was labeled with a number. Therefore, it is possible to name every surface clearly by combining slice letters and surface numbers (Figure 4).

In addition, since all gel samples were processed in different orientations, it was important to define the cutting procedure for each orientation and always follow the same procedure for all its replications. It can be seen in Figure 5 that for each horizontal orientation (H–90°–OFF, H–0°–OFF, H–135°–OFF, H–45°–OFF, H–ON, and V–ON), the top surface of the sample was assigned as A1 and the bottom surface was assigned as D5. However, for V–OFF orientation, the lateral surface in front of the microwave door was considered as A1 and the opposite lateral surface was assigned as D5 (see Figure 5).

In this orientation, the bottom of the sample seen in the posters represents the part of each sample surface that has contact with the turntable. For V–ON orientation, the sample and its lateral surfaces were rotated during microwave heat processing and thus one of the four lateral surfaces was randomly assigned as A1. Finally, after cutting the samples, the gel slices were cooled using a mixture of ice, water, and sodium chloride (NaCl) in a box covered with a thin plastic sheet to ensure that the gel samples did not absorb water. With this quick decline in temperature, any further Maillard reaction was stopped after the microwave heat processing.

### 2.4. Thermal Pictures

After the gellan gel samples were heated in the microwave systems for 2 min and before their cutting, thermal (IR) pictures were taken from their A1 surface (see Figure 5) with an infrared camera (FLIR E53, FLIR Systems Inc., Wilsonville, OR, USA). Afterward, the thermal pictures were edited with special software (FLIR tools) to make the picture more accessible for interpretation. Most importantly, the color scale was changed to a “rainbow color palette” ranging from 20 °C to 100 °C. In the following IR pictures, low temperatures are presented in purple and dark blue and high temperatures are presented in red and white. It is important to mention that IR pictures were not taken from DDI in the Teflon beakers during identifying parallel output power levels, which was described in Section 2.1. In this case, only correlation between absorbed and output powers for different sample orientations in both microwave systems are presented in Figure 6.

### 2.5. Preparation of Posters

There is a need to present the photographs and thermal pictures of each gellan gel sample in an easily accessible and understandable format. Therefore, posters containing photographs (Figure 7) or the combination of thermal pictures and photographs for the A1 surface (Figures 8 and 9), which provide an overview of different sample orientations, were created. To achieve this goal, one of three replications, which was considered the most representative of the Maillard color changes, was chosen to be displayed in the posters.

## 3. Results and Discussion

### 3.1. Absorbed Power Levels of Solid-State and Magnetron Microwave Systems

The absorbed power plotted against the output power can be seen in Figure 6A,B for the solid-state and magnetron microwave systems, respectively. Comparing both microwave systems in Figure 6A,B showed that the samples heated in different orientations in the solid-state microwave system absorbed more microwave energy than those in the magnetron microwave system. In this figure, the output power of 300 W is perfect for direct comparison of both microwave systems because it was the only similar output power level investigated for both microwave systems. For instance, the absorbed powers at 300 W output power in different orientations were in the range of 154 W to 177 W for the solid-state microwave system and in the range of 62 W to 135 W for the magnetron microwave system.

In addition, it can be seen in Figure 6A,B that the absorbed powers of the solid-state microwave system in different orientations were much more similar to those in the magnetron microwave system. In more detail, it can be seen in this figure that for the magnetron microwave system, the absorbed power levels in the vertical orientations (V–ON and V–OFF) were considerably higher than those in the horizontal orientations. More power absorbed by samples in the vertical positions in the magnetron microwave system (Figure 6A) can show different microwave field distributions in the horizontal and vertical orientations in the area in this microwave cavity between the turntable surface to a height of 6 cm.

It is worth mentioning that Table 1 shows that the equivalent output power levels in the solid-state microwave system for both vertical orientations (V–ON and V–OFF) were significantly different and higher compared to other equivalent output power levels in the horizontal orientations. This can be attributed to the considerably higher absorbed power levels of the magnetron microwave system in these two vertical orientations compared to the absorbed power levels of the horizontal orientations in this microwave system (Figure 6). Therefore, the non-uniform and different magnetron microwave heating for the vertical and horizontal orientations resulted in considerably different equivalent output power levels in the solid-state microwave system for similar vertical and horizontal orientations.

### 3.2. Homogeneity of Microwave Heating

For the evaluation of the homogeneity of microwave heating of samples with different orientations in the solid-state and magnetron microwave systems, the photographs of their slices were arranged in posters presented in Figure 7. The target of this step is to compare differences of the same orientations in the solid-state and magnetron microwave systems. It can be seen in Figure 7 that two orientations of H–ON and H–135°–OFF did not show or showed only very small brown spots in both microwave systems. In more detail, the most uniform color changes without any brown spots occurred in the H–ON treatment for both microwave systems, perhaps due to a more uniform microwave field distribution at the bottom of the microwave cavity for horizontal orientations and a better microwave power distribution provided by rotation of the turntable. In addition, the color change in orientation of H–135°–OFF was lower than other orientations, perhaps because this treatment absorbed the least power (61.8 W, see Table 1). It might also be due to a more uniform microwave energy distribution over the whole sample processed in this orientation.

The sample processed in H–0°–OFF orientation in the solid-state system also showed no highly visible brown spots; only one small brown spot was visible on the surface D5. In contrast, the same orientation in the magnetron microwave system caused dark spots in the top left corner for the surfaces B3, C4, and D5. This confirmed less uniform microwave heat distribution in this orientation for the magnetron microwave system than that in the solid-state system.

For H–45°–OFF and H–90°–OFF orientations, brown spots showed in all samples processed in both solid-state and magnetron microwave systems on surfaces C4 and D5, with the strongest browning on surface D5. Moreover, the browned areas in the samples processed in these two orientations were also larger and stronger in the magnetron system. In more detail, for H–45°–OFF orientation, the brown spot was mostly located on the top part of the samples processed in both microwave systems. The brown spot was again located at the top part of samples processed in the solid-state system for H–90°–OFF orientation. However, the browned area was on the right side of samples processed in the magnetron system in this orientation.

Figure 7 clearly shows that the browning patterns were very different between vertical and horizontal orientations in both microwave systems. The intensity of Maillard color changes in the vertical orientations was greater than those in the horizontal orientations. This is possibly due to these vertical orientations absorbing around two times more power than horizontal orientations (see Table 1). In addition, Figure 7 shows that the browning patterns for vertical orientations were very different between both microwave systems. While both orientations in the solid-state system resulted in a brown spot at the bottom of the samples, multiple browning spots can be found on all surfaces of samples processed in the magnetron system. In addition, the heating patterns in these two orientations in the magnetron system were different to each other. However, there is only a very small difference between the shape and intensity of brown spots in two vertical treatments (V–ON and V–OFF) in the solid-state system.

Overall, it can be concluded that the solid-state microwave system provides more uniform heating than the magnetron system in both vertical and horizontal orientations, especially at the H–ON orientation using the Maillard color changes. A higher uniformity of heating in the solid-state microwave system was found by our previous research investigating different positions [11], different sample shapes [16], and different frequencies [20]. Generally, uneven heating and formation of hot spots are famous problems for the heat processing of samples in the magnetron microwave system [2]. However, Figure 7 shows that although the microwave system type (magnetron or solid-state) has an obvious influence on heating uniformity in the sample, the sample orientation also has a large influence. This means that through the selection of the appropriate orientation of sample or appropriate package design, it is possible to provide uniform microwave heat processing even in the magnetron microwave system.

### 3.3. Assessment of Temperature Distribution by Infrared Absorption

The IR pictures and photographs of sample surface A1 for each orientation in both magnetron and solid-state microwave systems are presented in Figure 8. In some IR pictures, a rectangular, thin shape with lower temperature (usually blue color) is visible around the sample, which results from the Teflon dish (Figure 2) below the gellan gel samples.

It can be seen in the IR pictures (Figure 8) that the temperature distribution of the samples varied significantly by changing the orientations in both the magnetron and solid-state microwave systems. In addition, it can be seen that the temperature distribution of each orientation showed notable variations between the solid-state and magnetron microwave systems. In more detail, Figure 8 shows that the magnetron microwave system provided less uniform heat processing with a dark blue and purple cold spot for all horizontal orientations of H–90°–OFF, H–0°–OFF, H–135°–OFF, H–45°–OFF, and H–ON. For instance, the long right side of the samples processed in H–0°–OFF and H–135°–OFF orientations in both solid-state and magnetron microwave systems was heated more, and mostly red in color (Figure 8). However, the rest of the samples processed in these orientations were less uniform in the magnetron system, as their color ranged from yellow to dark blue (Figure 8). This figure clearly shows that besides the heated side of the samples processed in these two orientations in the solid-state microwave system, the rest of the samples appeared from yellow to light blue in color. Therefore, the heating was more uniform in the solid-state microwave system for these orientations. Similar results could be seen for the orientations of H–45°–OFF and H–90°–OFF because there was more temperature variation in these orientations for samples processed in the magnetron microwave system. In fact, in some parts of these samples with a dark blue color, it was almost as cold as the sample at the beginning of the heat processing (Figure 8). Therefore, the solid-state system offers more uniform heating in these horizontal orientations.

Figure 8 shows that the orientation H–ON in the solid-state microwave system provided the most uniform temperature distribution compared to other horizontal orientations in this microwave system, and also compared to all horizontal orientations in the magnetron microwave system. In more detail, the IR picture of the sample processed at orientation H–ON in the solid-state microwave system displayed approximately a hot ring in a red color forming at the sample borders, while the middle part was yellow. Therefore, the H–ON orientation in the solid-state microwave system showed the most uniform temperature distribution (Figure 8) and Maillard color changes (Figure 7) between all horizontal orientations in both microwave systems. Of note, in Figure 7, there were no brown spots visible for both magnetron and solid-state magnetron systems in this orientation. This suggests that the time–temperature history in the hot spots was not sufficient for the production of visible Maillard color changes.

Generally, both vertical orientations, even V–OFF orientation in the magnetron system, resulted in more homogenous temperature distribution than horizontal orientations in this microwave system. These two vertical orientations also resulted in homogenous temperature distribution and almost similar temperature distribution to the H–ON orientation in the solid-state microwave system. Homogenous temperature distribution of vertical orientations in both magnetron and solid-state microwave system can be referred to as almost double-aborted power levels in these orientations (see Table 1). In fact, as these two vertical orientations (V–ON and V–OFF) received more absorbed power, it can be concluded that it is possible to increase the uniformity of microwave temperature distribution by increasing the output power (this hypothesis will be investigated in the following section). However, the Maillard color changes of these vertical orientations, even V–ON orientation for both microwave systems, especially for the magnetron microwave system, were not very homogenous (Figure 7). Therefore, increasing the absorbed power can improve the temperature distribution, but not the temperature–time distribution of the sample.

### 3.4. Comparison of Different Absorbed Power Levels on Heating Uniformity

In fact, to solve the problem of very different absorbed power levels of horizontal and vertical positions and to compare all orientations of the solid-state microwave system with each other at similar absorbed power levels, the output power in these two vertical orientations was calculated based on the average absorbed power in all five horizontal treatments, as noted in Table 1. Therefore, in this part of the study, the effect of two levels of absorbed power on the heat uniformity of samples processed at vertical orientations was compared. In fact, using this average absorbed power (67.6 W) and related equations of solid-state microwave system in these orientations, new equivalent output power levels in this microwave system were calculated for this step (Table 2). Figure 9 shows that the reduction in absorbed power to 67.6 W for both V–ON and V–OFF orientations resulted in more uniform Maillard color changes. However, it resulted in a less uniform temperature distribution in IR pictures. Therefore, we can conclude that in similar absorbed power levels, the H–ON orientation resulted in the most uniform Maillard color changes, and simultaneously, the most uniform temperature distribution. In other words, it seems that the microwave field distribution in the horizontal direction is more uniform than the vertical direction in the solid-state microwave system.

## 4. Conclusions

In this study, the homogeneity of microwave heat distribution within cuboid food model samples was investigated in different orientations in the solid-state and magnetron microwave systems. In more detail, seven different orientations (H–0°–OFF, H–45°–OFF, H–90°–OFF, H–135°–OFF, H–ON, V–ON, and V–OFF) were defined, and the samples were processed for 120 s. The heating patterns were assessed on gellan gel samples containing Maillard reactants using the thermal pictures and photographs of the samples. For both microwave systems, the output power calibration and measurement of efficiency were carried out. After investigating the results, it was found that the solid-state microwave system had a similar and higher heating efficiency in all investigated orientations compared to the magnetron microwave system. However, the results showed that placing the samples in vertical positions in the magnetron microwave cavity allowed them to absorb more energy than horizontal orientations, and thus for more microwave energy efficiency, the vertical positions in the magnetron microwave system can be used. This also confirms the non-uniform microwave field distribution in horizontal and vertical directions in this microwave system. The highest uniformity was achieved with H–ON in the solid-state microwave system. This orientation led to more uniform Maillard color changes and temperature distribution than the others, even for the V–ON orientation at a similar absorbed power level. Because we investigated the horizontal and vertical orientations at the bottom of the microwave cavity in this paper, further research will investigate the microwave field distribution at different vertical levels in both microwave cavities using a whole meal containing both liquid and solid phases in the package at different sizes and shapes.

## Figures and Tables

**Figure 1 foods-10-01986-f001:**
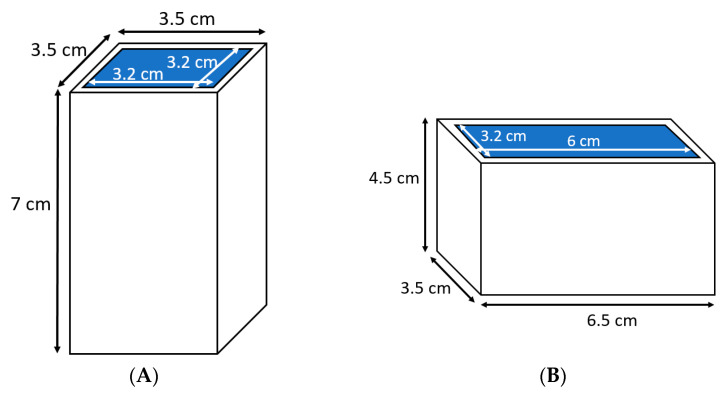
Two Teflon beakers used to measure the absorbed power of DDI water for (**A**) vertical and (**B**) horizontal orientations.

**Figure 2 foods-10-01986-f002:**
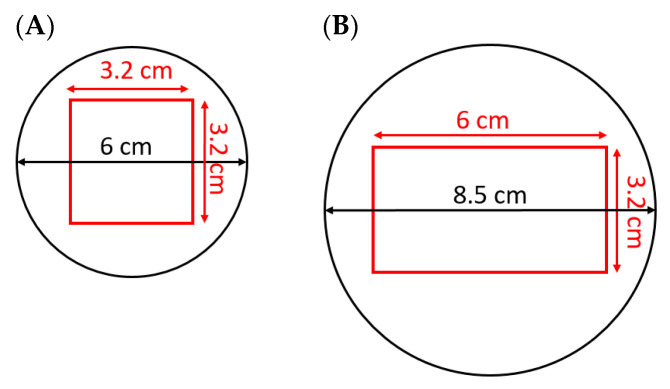
Teflon plates used under the gellan gel samples during microwave heat processing for (**A**) vertical and (**B**) horizontal orientations.

**Figure 4 foods-10-01986-f004:**
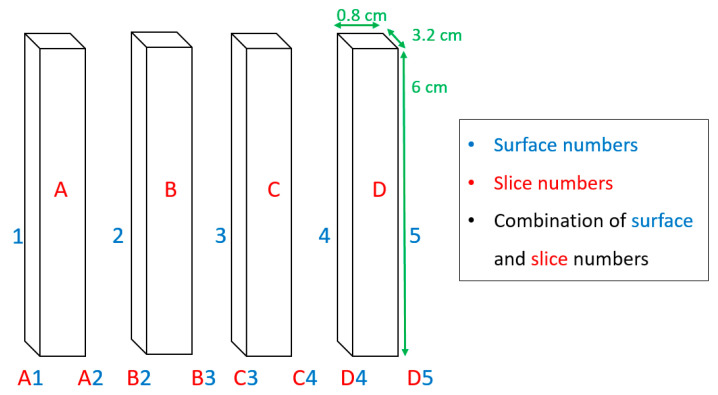
Surface and slice numbers of the gellan gel samples after cutting into four equal slices, called A, B, C, and D.

**Figure 5 foods-10-01986-f005:**
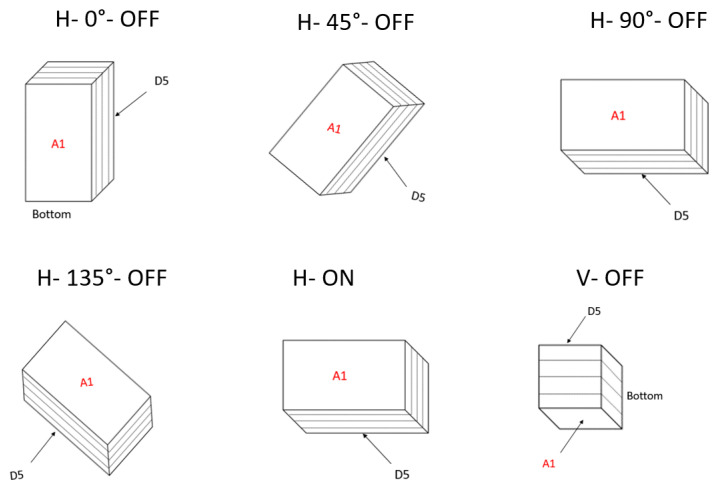
Top view of gellan gel samples in different orientations. The abbreviations of V, H, ON, and OFF represent vertical orientation, horizontal orientation, turntable on, and turntable off, respectively.

**Figure 6 foods-10-01986-f006:**
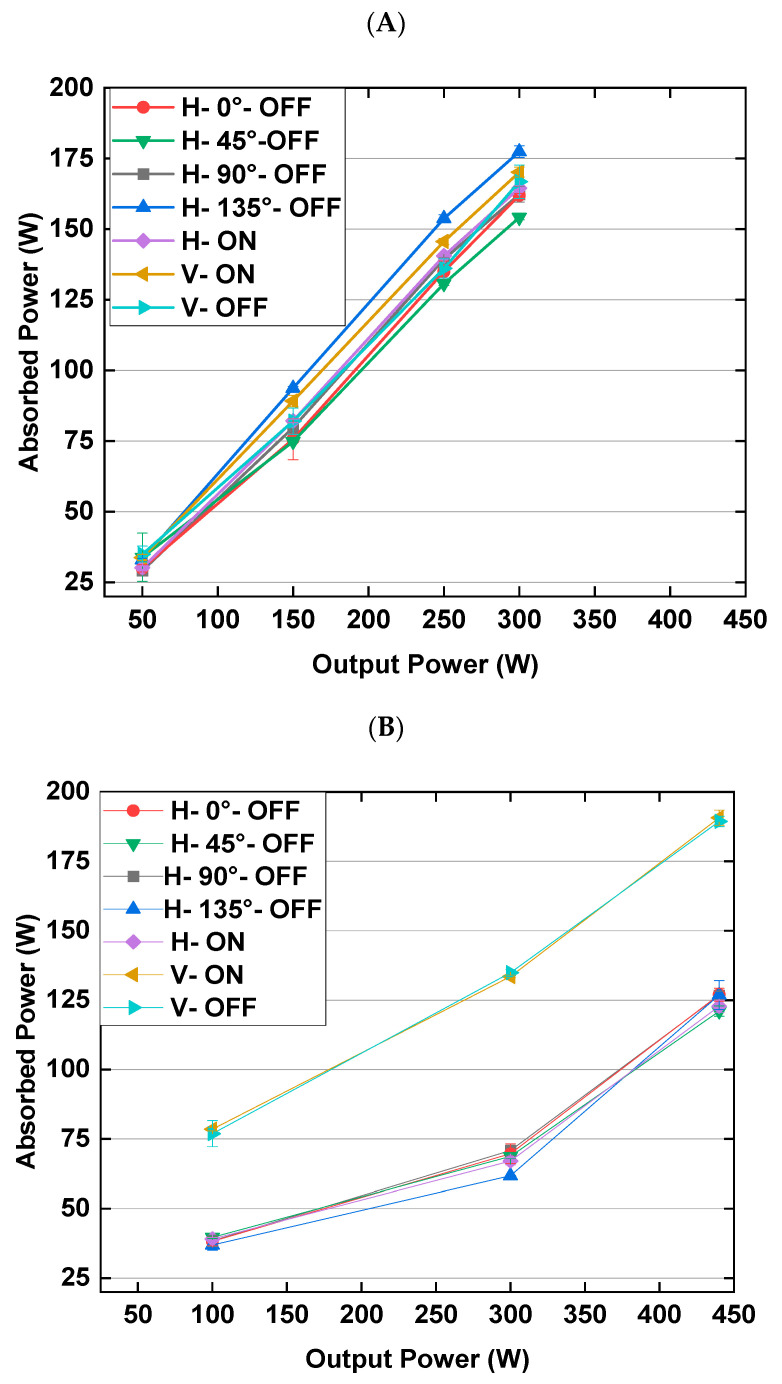
Correlation between absorbed and output powers for different sample orientations in (**A**) the solid-state microwave system and (**B**) magnetron microwave system.

**Figure 7 foods-10-01986-f007:**
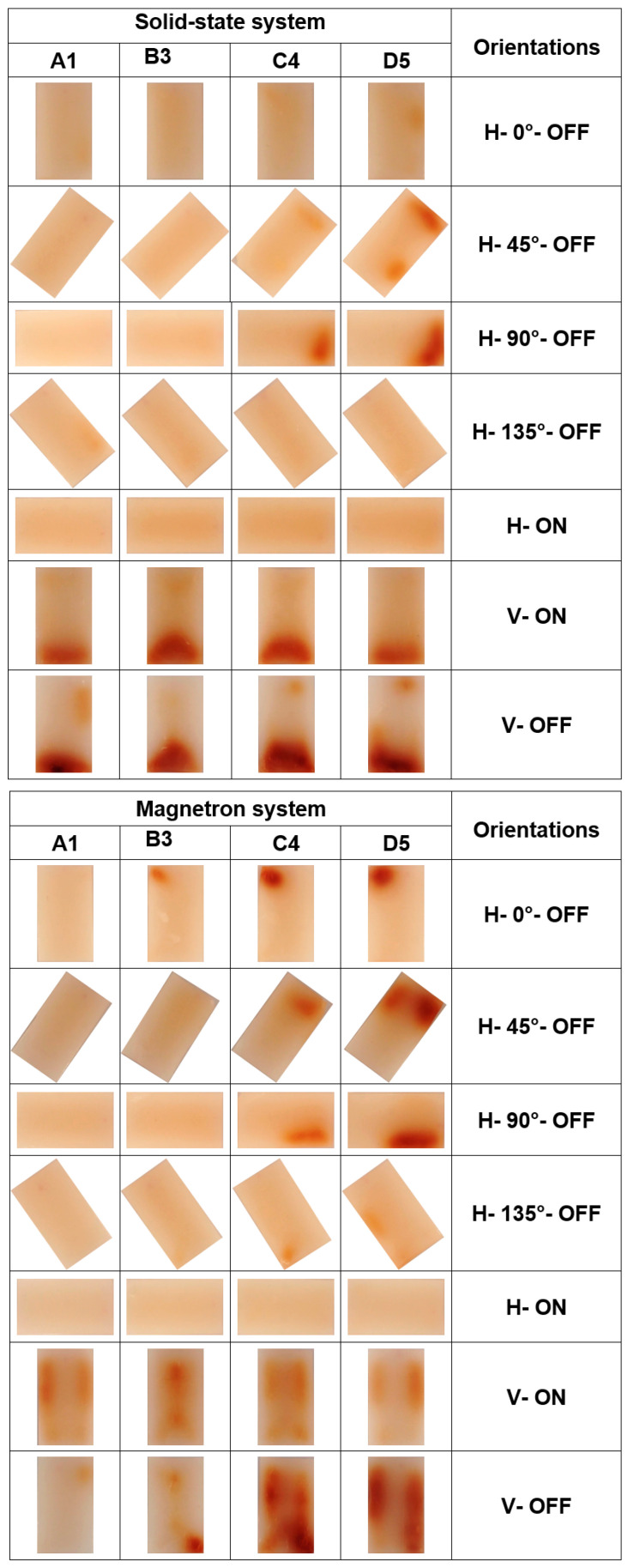
Photographs of sample surfaces A1, B3, C4, and D5 processed at different orientations after 120 s heat processing in the solid-state and magnetron microwave systems.

**Figure 8 foods-10-01986-f008:**
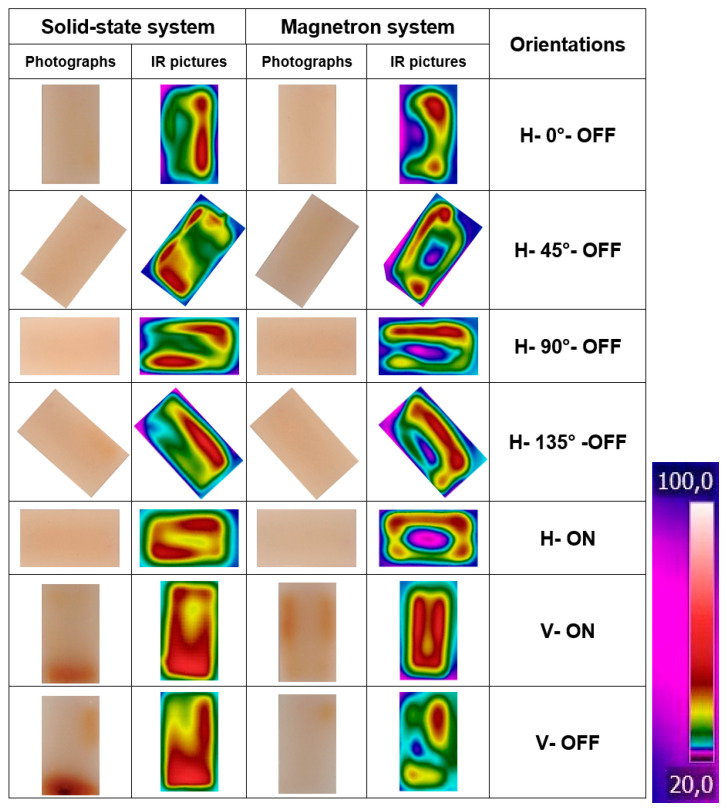
Thermal (IR) pictures and photographs of the A1 surface from gellan gel samples processed in the solid-state or magnetron microwave in different orientations. The temperature-color guideline is at right.

**Figure 9 foods-10-01986-f009:**
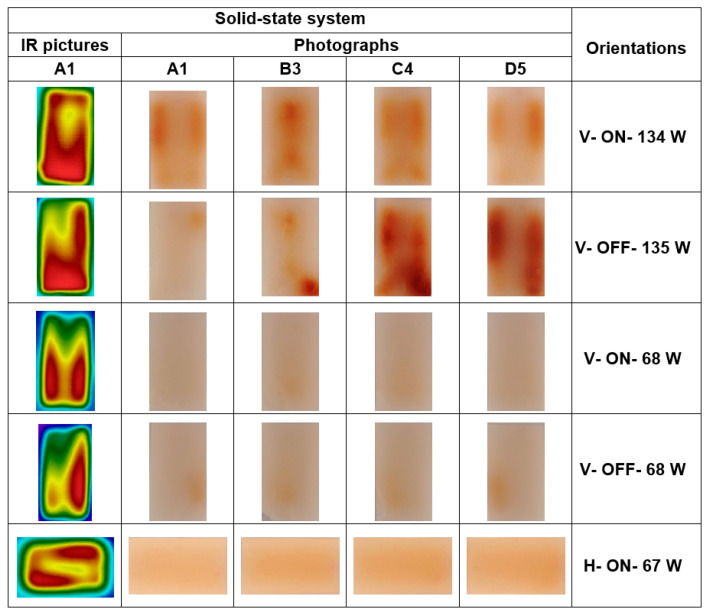
Photographs of sample surfaces A1, B3, C4, and D5 and the IR picture of surface A1 processed at vertical (V) and horizontal (H) orientations with (ON) and without (OFF) turntable rotation at different absorbed power levels after 120 s heat processing in the solid-state microwave systems.

**Table 1 foods-10-01986-t001:** Parallel output power levels for both solid-state and magnetron microwave systems for different orientations. In this table, the linear correlations between absorbed and output powers for different sample orientations in the solid-state microwave system.

Orientations	Linear Equations and R^2^ Values of the Solid-State System		Absorbed Power Levels (W) on Both Microwave Systems	Equivalent Output Power Levels (W) of the Solid-State System
H–0°–OFF	y = 0.5353x + 0.4310	R^2^ = 0.9966	69.68 ± 3.55	129
H–45°–OFF	y = 0.5146x + 0.2046	R^2^ = 0.9988	68.67 ± 0.82	133
H–90°–OFF	y = 0.5418x + 1.0599	R^2^ = 0.9984	70.83 ± 1.88	129
H–135°–OFF	y = 0.5836x + 5.0398	R^2^ = 0.9985	61.79 ± 0.10	97
H–ON	y = 0.5444x + 2.2912	R^2^ = 0.9991	67.05 ± 1.41	119
V–ON	y = 0.5496x + 6.6302	R^2^ = 0.9996	133.50 ± 0.42	231
V–OFF	y = 0.5260x + 6.3627	R^2^ = 0.9975	134.85 ± 1.06	244

**Table 2 foods-10-01986-t002:** Parallel output power levels for absorbed power level of 67.6 W in the vertical orientations in the solid-state microwave system.

Orientations	Linear Equations and R^2^ Values of the Solid-State System Extracted from Figure 6A		Average Absorbed Power Level (W) of All Five Horizontal Posi Shown in Table 1	Equivalent Output Power Levels (W) of the Solid-State System
V–ON	y = 0.5496x + 6.6302	R^2^ = 0.9996	67.6	111
V–OFF	y = 0.5260x + 6.3627	R^2^ = 0.9975	67.6	116

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
