# Peer review of "Effect of Vertical and Horizontal Sample Orientations on Uniformity of Microwave Heating Produced by Magnetron and Solid-State Generators"

_foods, 2021, doi:10.3390/foods10091986_

Round 1

Reviewer 1 Report

In this paper, the authors compared the microwave heating uniformity of magnetron and solid-state generators. There are several problems, ranging from the novelty and the experiment design. It is more like optimization of microwave heating instrument rather than the mechanism of food matrix influenced by microwave. The manuscript should be improved. In order to make easier the revision process a list of queries is stated below.

Hyphens are improperly over used in this manuscript.

Material and methods

Line 113-126; Lines 232-325: Different matrix may have different dielectric constants which lead to different microwave absorbance. Why double deionized water was chosen to identify the absorbed power, while the heating uniformity was determined based on gellan gel; what is the relationship between them (the matrix: water VS gel; the parameter: absorbed power VS heating uniformity)?

Line 146; Lines 327-405: What is “Maillard color changes”? Please clarify it. The mechanism of the color change caused by the Maillard reaction was not even mentioned in the manuscript. The photographs which were used to characterize the Maillard color change are hard to distinguish. More accurate technology can be taken into consideration in measuring the uniformity of microwave heating on gellan gel. For example, scanning electron microscope and texture analyzer can be used to characterize the micro- and macro- profile of the gel, and this is more creditable than photograph.

Author Response

Revision of manuscript entitled

"Effect of Vertical and Horizontal Sample Orientations on Uni‎formity of Microwave Heating Produced by Magnetron and ‎Solid-state Generators ‎"

submitted to journal of "Foods". 

Dear Editor and Reviewers,

Thank you very much for assessing our manuscript and for the useful comments and suggestions on the manuscript. We modified the manuscript accordingly and these changes in the manuscript were highlighted with yellow color. In addition, answers to your comments are listed below:

Editor and Reviewer comments:

Editor

Comment: We notice the format of reference did not meet the standard of our journal, so we attached the template of our journal, hope you could revise it according to the template.

Answer: The revision of references was done according to the format of journal.

Comment: In addition, The length of the present version is a little shorter than what we expect for article papers. In order to increase the readability of the article and to have a deeper understanding of the research content for readers, we are kindly suggesting you to add more references to support your research results when you revise your manuscript.

Answer: It was done.

Reviewer 1

Comments and Suggestions for Authors: In this paper, the authors compared the microwave heating uniformity of magnetron and solid-state generators. There are several problems, ranging from the novelty and the experiment design. It is more like optimization of microwave heating instrument rather than the mechanism of food matrix influenced by microwave. The manuscript should be improved. In order to make easier the revision process a list of queries is stated below.

Answer: It can be helpful to mention here that the target of this study was investigation of the effect of sample orientations on uni‎formity of microwave heating produced by magnetron and ‎solid-state generators. We did not want to investigate the mechanism of food matrix influenced by microwave ‎as it was done before by other researchers. The targets of the manuscriot are mentioned in the introduction.

Hyphens are improperly over used in this manuscript.

Answer: It was corrected.

Material and methods

Line 113-126; Lines 232-325: Different matrix may have different dielectric constants which lead to different microwave absorbance. Why double deionized water was chosen to identify the absorbed power, while the heating uniformity was determined based on gellan gel; what is the relationship between them (the matrix: water VS gel; the parameter: absorbed power VS heating uniformity)?

Answer: More information was added in section 2.1 of the manuscript to answer your question.

Line 146; Lines 327-405: What is “Maillard color changes”? Please clarify it. The mechanism of the color change caused by the Maillard reaction was not even mentioned in the manuscript. The photographs which were used to characterize the Maillard color change are hard to distinguish. More accurate technology can be taken into consideration in measuring the uniformity of microwave heating on gellan gel. For example, scanning electron microscope and texture analyzer can be used to characterize the micro- and macro- profile of the gel, and this is more creditable than photograph.

Answer: The clarification of Maillard reaction and Maillard color changes were added in the last paragraph of introduction.

We and a lot of other researchers using Maillard reaction for investigation of heat processing believe that the brown color changes resulted from Maillard reaction can easily and perfectly be seen in the samples (For instance in Figures 7 and 8). Please consider this note that the target for application of Maillard indicator is avoiding the application of time-consuming and expensive procedures (even result in some cases not very accurate and repeatable data) to investigate micro- and macro- profile changes of samples.

Thank you and best regards,

Somayeh Taghian Dinani

Reviewer 2 Report

The method for measuring the average microwave power delivered to the different specimens is good.

I also like the model system approach to measuring Maillard reactions in situ.

Major comments:

However, the non-uniformity of the electromagnetic fields in microwave cavities is well known and thoroughly understood through both modelling and experimental studies. Therefore the orientation of the food samples will inevitably affect the exposure of foods to different microwave intensities.

The study presented in the paper is therefore unsurprising and a very practical set of outcomes that are symptomatic of the non-uniformity of the electromagnetic fields in the microwave cavity.

The claimed novelty of the paper is the association of the orientation of the food, the occurrence and effects of Maillard reactions and the non-uniformity of the fields.

Needs to be clarified: how were the power levels in the microwave systems modulated when less than 100%? Continuously, or on-off? This difference is significant in terms of hot spots and overheating.

How did the Teflon plates (lines 154-162) affect the microwave field distribution compared with the situation where there were no Teflon plates in the cavity?

How were the moisture contents of the gel samples measured experimentally in-situ, and how uniform were the moisture contents inside each gel sample?

The homogeneity of the heating is discussed in section 3.2, but the homogeneity of the moisture contents does not appear to be discussed anywhere, and such a discussion about the homogeneity of the moisture contents inside the samples is necessary.

Maillard reactions are chemical reactions, and the reaction rates depend on the reactant concentrations, the product temperatures and the product moisture contents. There is no modelling in this paper, and the results seem to have limited value, because the product moisture contents (as inputs) are not reported in the context of the results (outputs).

Minor comments:

Lines 70, 77, 153:            “micro-wave” should be “microwave”

Line 81: “uniformi-ty” should be “uniformity”

Line 308: “brown-ing” should be “browning”

Line 380: “ho-mogeneous” should be “homogeneous”

Author Response

Revision of manuscript entitled

"Effect of Vertical and Horizontal Sample Orientations on Uni‎formity of Microwave Heating Produced by Magnetron and ‎Solid-state Generators ‎"

submitted to journal of "Foods". 

Dear Editor and Reviewers,

Thank you very much for assessing our manuscript and for the useful comments and suggestions on the manuscript. We modified the manuscript accordingly and these changes in the manuscript were highlighted with yellow color. In addition, answers to your comments are listed below:

Editor and Reviewer comments:

Editor

Comment: We notice the format of reference did not meet the standard of our journal, so we attached the template of our journal, hope you could revise it according to the template.

Answer: The revision of references was done according to the format of journal.

Comment: In addition, The length of the present version is a little shorter than what we expect for article papers. In order to increase the readability of the article and to have a deeper understanding of the research content for readers, we are kindly suggesting you to add more references to support your research results when you revise your manuscript.

Answer: It was done.

Reviewer 2

Comments and Suggestions for Authors: The method for measuring the average microwave power delivered to the different specimens is good. I also like the model system approach to measuring Maillard reactions in situ.

Answer: Thank you very much.

Major comments:

However, the non-uniformity of the electromagnetic fields in microwave cavities is well known and thoroughly understood through both modelling and experimental studies. Therefore the orientation of the food samples will inevitably affect the exposure of foods to different microwave intensities.

The study presented in the paper is therefore unsurprising and a very practical set of outcomes that are symptomatic of the non-uniformity of the electromagnetic fields in the microwave cavity.

The claimed novelty of the paper is the association of the orientation of the food, the occurrence and effects of Maillard reactions and the non-uniformity of the fields.

Answer: We do not agree with this comment. Solid-state microwave system can provide a much uniform heat processing than magnetron microwave system. However, in this paper, we showed that the unevenness of microwave power absorption and heating in both solid-state and magnetron systems is not predictable and requires checking with specialized devices, such as using food model system containing Maillard reaction. In other words, although the microwave system type (magnetron or ‎solid-state) has an obvious influence on heating uniformity in the sample, the sample ‎orientation also has a large influence. This means that by the selection of appropriate ‎orientation of sample or appropriate package design, it is possible to provide a uniform ‎microwave heat processing even in the magnetron microwave system.‎

Needs to be clarified: how were the power levels in the microwave systems modulated when less than 100%? Continuously, or on-off? This difference is significant in terms of hot spots and overheating.

Answer: The answer of your question was added in the third paragraph of introduction. 

How did the Teflon plates (lines 154-162) affect the microwave field distribution compared with the situation where there were no Teflon plates in the cavity?

Answer: The answer of this question is added in section 2.3 of the manuscript.

How were the moisture contents of the gel samples measured experimentally in-situ, and how uniform were the moisture contents inside each gel sample?

Answer: The moisture content of samples after heat processing was not measured. In fact, as one of the target of this study was 3D assessment of heating profile of samples, the time-energy consuming and non-accurate procedure of measuring moisture content of samples cannot provide special information in this case.

The homogeneity of the heating is discussed in section 3.2, but the homogeneity of the moisture contents does not appear to be discussed anywhere, and such a discussion about the homogeneity of the moisture contents inside the samples is necessary.

Maillard reactions are chemical reactions, and the reaction rates depend on the reactant concentrations, the product temperatures and the product moisture contents. There is no modelling in this paper, and the results seem to have limited value, because the product moisture contents (as inputs) are not reported in the context of the results (outputs).

Answer: Please consider this note that the target of this paper was not modelling and it was comparison of different orientations in the solid-state and magnetron microwave system. Thus, a homogenous and consistent model sample containing Maillard reaction substrates was designed and used for all treatments and the color changes were used for a three dimensional visualization of the heating pattern of samples. In other words, for all treatments, similar and uniform samples containing Maillard substrates were used. In fact, the time-energy consuming and non-accurate procedure of measuring moisture content of samples cannot provide special information about 3D heat processing of samples. In addition, Maillard color changes can also show the time-temperature history of sample. More information was provided in the introduction and section 2.2 of the manuscript in this issue.

Minor comments:

Lines 70, 77, 153:  “micro-wave” should be “microwave”

Answer: It was done.

Line 81: “uniformi-ty” should be “uniformity”

Answer: It was done.

Line 308: “brown-ing” should be “browning”

Answer: It was done.

Line 380: “ho-mogeneous” should be “homogeneous”

Answer: It was done.

Thank you and best regards,

Somayeh Taghian Dinani

Reviewer 3 Report

Research is very important; it has an important scientific and practical dimension. The purposefulness of the conducted study was demonstrated and justified. The uniformity of food heating with microwaves is extremely important. The presented results can be very useful for the construction of suitable microwave devices on a larger scale. It turns out that the unevenness of microwave power absorption and heating in both semiconductor and magnetron systems is not predictable and requires checking with specialized devices, such as in this manuscript. Therefore, this research is very valuable. The authors made sure that the tests were carried out correctly in terms of obtaining the same absorbed power in both microwave systems. They interpreted and discussed the obtained results well, although there is no literature on this subject.

The manuscript was prepared properly, well written and organized. I have no comments to the title, keywords, introduction, materials and methods, experimental procedures, references (all relevant, but rather inadequately cited without numbering, or editorial requirements have changed ..).

The results were presented correctly, all figures and tables were cited. The results are discussed and the conclusions are supported by the evidence. The conclusions and abstracts could further highlight the relevance and the need for further research, for example on how the samples are distributed and sized, and depending on the food moisture and other properties.

Some other comments:

Line 157: It is not clearly written how the samples were positioned at different angles on plates with 2 mm deep.? What is this deep for?

Lines 164-165: the frequency unit should be the same, either MHz or GHz, the values are the same for both techniques, so why are they in different units? Also, correct this sentence stylistically.

In addition, in many places the text must be edited, for example, remove unnecessary "-" probably resulting from splitting words before converting to pdf, e.g. lines 77, 81, 112.

What are the linear equations in Tab. 2? Do they relate to Fig. 6? From this, the conclusion is that the captions of tables and figures are not sufficiently informed about what they contain.

Author Response

Revision of manuscript entitled

"Effect of Vertical and Horizontal Sample Orientations on Uni‎formity of Microwave Heating Produced by Magnetron and ‎Solid-state Generators ‎"

submitted to journal of "Foods".

Dear Editor and Reviewers,

Thank you very much for assessing our manuscript and for the useful comments and suggestions on the manuscript. We modified the manuscript accordingly and these changes in the manuscript were highlighted with yellow color. In addition, answers to your comments are listed below:

Editor and Reviewer comments:

Editor

Comment: We notice the format of reference did not meet the standard of our journal, so we attached the template of our journal, hope you could revise it according to the template.

Answer: The revision of references was done according to the format of journal.

Comment: In addition, The length of the present version is a little shorter than what we expect for article papers. In order to increase the readability of the article and to have a deeper understanding of the research content for readers, we are kindly suggesting you to add more references to support your research results when you revise your manuscript.

Answer: It was done. 

Reviewer 3

Comments and Suggestions for Authors: Research is very important; it has an important scientific and practical dimension. The purposefulness of the conducted study was demonstrated and justified. The uniformity of food heating with microwaves is extremely important. The presented results can be very useful for the construction of suitable microwave devices on a larger scale. It turns out that the unevenness of microwave power absorption and heating in both semiconductor and magnetron systems is not predictable and requires checking with specialized devices, such as in this manuscript. Therefore, this research is very valuable. The authors made sure that the tests were carried out correctly in terms of obtaining the same absorbed power in both microwave systems. They interpreted and discussed the obtained results well, although there is no literature on this subject.

Answer: Thank you very much.

The manuscript was prepared properly, well written and organized. I have no comments to the title, keywords, introduction, materials and methods, experimental procedures, references (all relevant, but rather inadequately cited without numbering, or editorial requirements have changed ..).

Answer: The format of references was modified.

The results were presented correctly, all figures and tables were cited. The results are discussed and the conclusions are supported by the evidence. The conclusions and abstracts could further highlight the relevance and the need for further research, for example on how the samples are distributed and sized, and depending on the food moisture and other properties.

Answer: It was done.

Some other comments:

Line 157: It is not clearly written how the samples were positioned at different angles on plates with 2 mm deep.? What is this deep for?

Answer: The cavities inside Teflon plates were designed to put samples on the Teflon plates at a similar place and ‎at their ‎center all the time (Figure 2). ‎Therefore, they were not designed for setting the samples at different angles. Figure 3 shows that for positioning the horizontal samples at different angles, the big Teflon plate (shown in Figure 2B) containing the sample in its center could be easily rotated.

Lines 164-165: the frequency unit should be the same, either MHz or GHz, the values are the same for both techniques, so why are they in different units? Also, correct this sentence stylistically.

Answer: The GHz unit was used for both cases. The sentence also was modified.

In addition, in many places the text must be edited, for example, remove unnecessary "-" probably resulting from splitting words before converting to pdf, e.g. lines 77, 81, 112.

Answer: It was done.

What are the linear equations in Tab. 2? Do they relate to Fig. 6? From this, the conclusion is that the captions of tables and figures are not sufficiently informed about what they contain.

Answer: The equations in column second of Table 2 and Table 1 are in linear format (Y=aX+b). Yes, these equations are related to the correlation between absorbed and output powers for different sample orientations in the solid-state microwave system in Figure 6 A. To clarify it better, the captions and column titles of Table 1 and Table 2 were modified.

Thank you and best regards,

Somayeh Taghian Dinani

Round 2

Reviewer 1 Report

Authors have satisfactorily addressed the comments/questions raised.

Author Response

Revision of manuscript entitled "Effect of Vertical and Horizontal Sample Orientations on Uni‎formity of Microwave Heating Produced by Magnetron and ‎Solid-state Generators‎" submitted to journal of "Foods".

Dear Editor and Reviewers,

Thank you very much for assessing our manuscript. Answers to your comments are listed below:

Reviewer 1

Authors have satisfactorily addressed the comments/questions raised.

Answer: Thank you very much. 

Thank you and best regards,

Somayeh Taghian Dinani

Reviewer 2 Report

The author response of “Please consider this note that the target of this paper was not modelling ” is not really satisfactory, because the data in this paper should be capable of being modelled, and such modelling is impossible without knowing the moisture contents. In other words, the experimental data in the paper for the Maillard reactions cannot be reproduced by any type of mathematical modelling without knowing the moisture contents, so the value of the data is diminished by this omission.

Author Response

Revision of manuscript entitled "Effect of Vertical and Horizontal Sample Orientations on Uni‎formity of Microwave Heating Produced by Magnetron and ‎Solid-state Generators‎" submitted to journal of "Foods".

Dear Editor and Reviewers,

Thank you very much for assessing our manuscript. Answers to your comments are listed below:  

Reviewer 1

Authors have satisfactorily addressed the comments/questions raised.

Answer: Thank you very much.

Reviewer 2

The author response of “Please consider this note that the target of this paper was not modelling ” is not really satisfactory, because the data in this paper should be capable of being modelled, and such modelling is impossible without knowing the moisture contents. In other words, the experimental data in the paper for the Maillard reactions cannot be reproduced by any type of mathematical modelling without knowing the moisture contents, so the value of the data is diminished by this omission.

Answer:

  • Modelling is not a mandatorial element of scientific publications. Modelling is helpful if experimental methods would not allow access to certain data/properties. But the Maillard reaction method is providing integral and spatially resolved data on the impact of microwave heating uniformity. Hence, why would modelling adding value? Methods of modelling have their own limitations, but why not indicate that the data presented here could be used by anyone to validate their modelling attempts.
  • Before being able to conduct and validate modelling, the methods to obtain the data must be developed/obtained
  • Materials and Methods are given in detail such that reproducing our experiments is possible in our opinion.
  • Considering moisture content of samples, the initial moisture content of gels (93.9 %) was already mentioned in section 2.2. In addition, as it was mentioned before for answering your question, a homogenous and consistent model sample containing Maillard reaction substrates, water, etc. was designed and used for all experiments.
  • Reducing water content is not a good factor in deciding about microwave heat processing and temperature distribution in the sample because of following reasons:
  • If we measure the moisture content in the whole sample after heat processing, no special conclusion can be obtained.
  • If we want to cut samples to some slices (for instance 4 slices similar to Figure 4), again the average moisture content of each slice cannot show the differences between the top and bottom of each slice as well as the right and left sides of each slice. However, the Maillard color changes can clearly and properly show these differences.
  • At high temperatures, in some parts of the sample, some minor or major tissue explosions and destruction can occur, and they can be used by water as escape paths. Thus, again, the temperature cannot be properly and correctly correlated with moisture content.
  • At the bottom of the sample, the water evaporation is restricted by the turntable. In fact, although the bottom of the sample has higher Maillard color changes and higher temperature, the moisture content is not reduced proportionally. Thus, again, the temperature cannot be properly and correctly correlated with moisture content.
  • During the cutting sample, we have moisture loss and, thus, measuring the moisture content of slices has a considerable error due to the nature of the test and also due to the water loss during cutting.

Best regards,

Somayeh Taghian Dinani